# Anti-Obesity Effect of *DKB-117* through the Inhibition of Pancreatic Lipase and *α-*Amylase Activity

**DOI:** 10.3390/nu12103053

**Published:** 2020-10-06

**Authors:** Do Hoon Kim, Yu Hwa Park, Jung Suk Lee, Hyun Il Jeong, Kye Wan Lee, Tong Ho Kang

**Affiliations:** 1Department of Oriental Medicine Biotechnology, College of Life Sciences and Graduate School of Biotechnology, Kyung Hee University, Global Campus, Gyeonggi 17104, Korea; hoony3914@gmail.com (D.H.K.); best1rd@gmail.com (Y.H.P.); 2R&D Center, Dongkook Pharm. Co., Ltd., Gyeonggi 16229, Korea; ljs@dkpharm.co.kr (J.S.L.); jhi@dkpharm.co.kr (H.I.J.); lkw1@dkpharm.co.kr (K.W.L.)

**Keywords:** *Phaseolus multiflorus* var. albus Bailey (PM), *Pleurotus eryngii* var. ferulae (PF), anti-obesity, pancreatic lipase, α-amylase

## Abstract

This study sought to evaluate the effects of *Phaseolus multiflorus* var. albus Bailey extract (PM extract) and *Pleurotus eryngii* var. ferulae extract (PF extract) on the inhibition of digestive enzymes and to confirm the anti-obesity effect of DKB-117 (a mixture of PM extract and PF extract) in digestive enzyme inhibition in a mouse model of obesity induced by a high-fat diet. In in vitro studies, PM extract and PF extract have increased dose-dependent inhibitory activity on *α-*amylase (Inhibitory concentration (IC_50_ value: 6.13 mg/mL)) and pancreatic lipase (IC_50_ value; 1.68 mg/mL), respectively. High-fat diet–induced obese mice were orally administered DKB-117 extracts at concentrations of 100, 200, and 300 mg/kg/day, while a positive control group was given orlistat (pancreatic lipase inhibitor) and *Garcinia cambogia* (inhibiting the enzymes needed to synthesize carbohydrates into fat) at concentrations of 40 and 200 mg/kg/day, respectively, for eight weeks. As a result, body weight, fat mass (total fat mass, abdominal fat, and subcutaneous fat) detected with microcomputed tomography, fat mass (abdominal fat and inguinal fat) after an autopsy, and liver triglyceride levels were decreased significantly in the DKB-117 (300 mg/kg/day) group compared to those in the HFD control group. Additionally, we obtained results indicating that the presence of carbohydrates was found more in the DKB-117-300 (300 mg/kg/day) group than in the HFD control group. These data clearly show that DKB-117 extracts are expected to have an anti-obesity effect through a complex mechanism that promotes carbohydrate release through the inhibition of carbohydrate-degrading enzymes while blocking lipid absorption through lipase inhibition.

## 1. Introduction

Obesity is a metabolic disorder characterized by an excess accumulation of fat in the body due to one’s energy intake exceeding one’s energy expenditure [1]. Obesity is a very common global health problem. It was reported by the World Health Organization in 2016 that more than 1.9 billion adults were overweight and, of these, more than 650 million were obese [2]. Obesity is a risk factor for metabolic syndrome and can lead to hypertension, type 2 diabetes (T2DM), dyslipidemia, cardiovascular disease (CVD), and stroke [3,4].

Currently, four weight-loss medicines (orlistat marketed as Xenical^®^, Roche Holding AG, Basel Switzerland or Alli^®^, GlaxoSmithKline, Brentford, UK; Contrave^®^ from Nalpropion Pharmaceuticals, San Diego, CA, USA; Belviq^®^ from Eisai, Tokyo, Japan; and Qsymia^®^ from Vivus, Campbell, CA, USA) have been approved by the United States Food and Drug Administration, with users’ body weight regulated by pancreatic lipase inhibition, increased energy consumption, and appetite suppression [5,6]. However, existing synthetic drugs have been reported to cause heart attack and stroke as well as liver damage [7]. 

Accordingly, there have been a number of studies conducted on the development of anti-obesity materials that are effective in decreasing appetite and reducing weight by using natural substances with long histories of use [8,9]. According to a recent report, many herbal extracts (*Garcinia cambogia*, *Plantago psyllium*, *Morus alba*) have been suggested to act on fat and carbohydrate metabolism to regulate body weight [10].

However, a lot of safety issues related to hepatic insufficiency, hepatitis, and heart disease have been reported in the case of *Garcinia cambogia* extract, which is the most widely sold and health functional food material in the world. Accordingly, it is required that new natural materials with good safety and efficacy be developed.

White kidney beans (*Phaseolus multiflorus* var. albus Bailey; PM) are native to Italy and belong to the Leguminosae family. Abdulwahid et al. reported that the white kidney bean–treated group in a diabetic-induced mouse model showed glucose and weight loss effects compared to the control group [10]. In addition, many studies on anti-obesity-related clinical trials have reported that white kidney bean ingestion is effective in weight loss, waist circumference reduction, and weight loss through α-amylase [11].

*Pleurotus eryngii* var. ferulae (PF) is a mushroom of the family Pleurotaceae that is rich in protein and dietary fiber [12]. Wang et al. and Alam et al. reported on the antioxidative, anti-inflammatory, and hypotensive effects of PF. In addition, PF water extracts reduced body weight, white adipose tissue weight, and liver weight in a mouse model of obesity induced by a high-fat diet (HFD) while improving glucose tolerance [1,13,14].

One of the most important strategies in the treatment of obesity includes the development of nutrient digestion and absorption inhibitors in an attempt to reduce the degree of energy intake through gastrointestinal mechanisms without altering any central mechanisms. The inhibition of digestive enzymes is one of the most widely studied mechanisms used to determine the potential efficacy of natural products as anti-obesity agents [15]. 

This study was carried out to develop a dietary supplement ingredient that improves convenience of use and has good safety and obesity effects by respectively mixing natural plants having α-amylase inhibitory and pancreatic lipase inhibitory effects. In the literature, *α*-amylase inhibitors are well documented to be effective in reducing postprandial hyperglycemia by slowing the digestion of carbohydrates and absorbing postprandial glucose [10]. Reducing postprandial hyperglycemia prevents glucose uptake into adipose tissue to inhibit the synthesis and accumulation of triacylglycerol. Lipase is a hydrolytic enzyme from the pancreas that changes triglycerides (TGs) to glycerol and fatty acids. Thus, the inhibition of lipase has an important role in the treatment of obesity by inhibiting fat absorption. Previously reported studies have shown the α-amylase–inhibitory effect of PM and the pancreatic lipase–inhibitory effect of PF [12,13].

Research is actively underway to find effective anti-obesity drugs or anti-obesity health functions. The authors of this study want to confirm the anti-obesity effect by mixing *Phaseolus multiflorus* var. albus Bailey with α-amylase inhibitory effect and *Pleurotus eryngii* var. ferulae reported to have a pancreatic lipase inhibitory effect. In this study, the optimum mixing ratio of PF extract and PF extract was selected by an in vitro test, and we sought to determine the anti-obesity effect of DKB-117 in digestive enzyme inhibition in a mouse model of obesity induced by HFD.

## 2. Materials and Methods

### 2.1. Plant Material Collection and Extract Preparation

DKB-117, which is a mixture of PM extract and PF extract, was provided by Dongkook Pharm. Co., Ltd. (Suwon, Korea). The lot number was DKB-117. PM (*Phaseolus multiflorus* var. albus Bailey) used in this study was cultivated from Egypt and purchased through Solim trading Co., Ltd. in Korea. PF (*Pleurotus eryngii* var. ferulae) was obtained from the DDLE A CHE Co., Ltd. (Cheonan, Korea). A voucher specimen number (DK0117) has been deposited at the R&D Center, Dongkook Pharm. Co., Ltd.

PM and PF were cut into small pieces and extracted with a 5-fold volume of 30% ethanol (v/v) at 80 °C for 10 h, in order to obtain two ethanolic extracts, namely PM and PF, respectively. After extraction, each solution was concentrated on a rotary evaporator (BUCHI Labortechnik AG, Flawil, Switzerland) until constant weight, and the PM extract was dried by freeze-drying and the PF extract was dried using a spray dryer. DKB-117 was prepared by mixing the previously prepared PM extract and PF extract at a weight ratio of 3:1.

### 2.2. α-Amylase Inhibition Assay

The measurement of α-amylase inhibitory activity was carried out via the iodine reaction method as described by Wilson et al. with a slight modification [16]. Briefly, α-amylase (Sigma-Aldrich, St. Louis, MO, USA) derived from human saliva was dissolved in phosphate-buffered saline (PBS) at a concentration of 20 unit/mL. As a substrate for α-amylase, soluble starch was dissolved in PBS at a concentration of 1%. To measure the inhibitory activity against α-amylase, 290 μL of PBS, 10 μL of α-amylase solution (20 unit/mL), and 50 μL of the test substance were mixed and preincubation was performed at 37 °C for 10 min. After preincubation, 350 μL of 1% soluble starch as substrate was added and reacted at 37 °C for 30 min. To determine the amount of soluble starch remaining after the reaction, 300 μL of iodine solution (0.1% KI + 0.01% I_2_/0.05 N HCl) was added to the reaction solution and the absorbance was measured at 620 nm using an enzyme-linked immunosorbent assay (ELISA) reader (Infinite 200 Pro; Tecan Austria GmBH, Grödig, Austria).

### 2.3. Pancreatic Lipase Inhibition Assay

Pancreatic lipase inhibitory activity was measured using the substrate p-nitrophenyl butyrate (PNPB) as described by Eom et al. with slight modification [17]. Briefly, an enzyme buffer was prepared by adding 30 μL of porcine pancreatic lipase (Sigma-Aldrich, St. Louis, MO, USA) in 10 mM of morpholinepropane sulfonic acid and 1 mM of ethylene diamine tetra acetic acid (pH: 6.8) to 850 mL of Tris buffer (100 mM of Tris-HCl and 5 mM of CaCl_2_; pH: 7.0). Then, 100 μL of DKB-117 or orlistat was mixed with 880 mL of the enzyme buffer and incubated for 15 min at 37 °C. After incubation, we added 20 μL of the substrate solution (10 mM of PNPB in dimethyl formamide) and the enzymatic reactions were allowed to proceed for 30 min at 37 °C. Pancreatic lipase inhibitory activity was determined by measuring the hydrolysis of PNPB to p-nitrophenol at 405 nm with the use of an ELISA reader (Infinite 200 Pro; Tecan Austria GmBH, Grödig, Austria). The activities of the negative control were reviewed with and without the inhibitor. The inhibitory activity (%) was calculated according to the formula below:Lipase inhibition (%) = [1 − (B − b)/(A − a)] × 100
where A is the activity of the enzyme without the inhibitor, a is the negative control without the inhibitor, B is the activity of the enzyme with the inhibitor, and b is the negative control with the inhibitor.

### 2.4. Animal Experiments

The protocol for animal study was approved by the Department of Biofood Research, KNOTUS Life Science Co., Ltd. Animal Ethics Committee (17-KE-265). Male C57BL/ JJmsSlc mice, five weeks old (18–20 g), were purchased from Central Lab Animal Inc. (Seoul, Republic of Korea). The animals were housed in polycarbonate cages (less than five mice/cage) under controlled temperatures (23 ± 3 °C), relative humidity (55% ± 5%), and lighting conditions (lights on from 08:00 to 20:00 h), with food and water made available ad libitum. Mice were fed either a regular diet (ND group; lab rodent chow; Cargill-Agri Purina, Seongnam-Si, Republic of Korea) or a 60% HFD (HFD group; Saeronbio, Uiwang-si, Republic of Korea). On the 19th day of HFD feeding, the body weight of the HFD group was 10% higher than that of the ND group. At this time, the HFD group was divided into six additional groups: an HFD control group, a DKB-117-100-treated group (100 mg/kg/day), a DKB-117-200-treated group (200 mg/kg/day), a DKB-117-300-treated group (300 mg/kg/day), an orlistat-treated group (40 mg/kg/day), and a *Garcinia cambogia*–treated group (200 mg/kg/day), respectively. For eight weeks, the mice were daily treated with test article and HFD, with the exception of the ND group.

After eight weeks of treatment, the mice were not fed for 15 h and their body weight was measured. Then, the mice were anesthetized with ether. Blood was drawn from the postcaval vein for serum biochemical analysis. Extracted liver, abdominal fat, and inguinal fat tissue were weighed.

### 2.5. Blood Analysis

Separated serum was examined for total cholesterol (TCHO), triglycerides (TG), high-density lipoprotein (HDL), and low-density lipoprotein (LDL) levels using a blood biochemical analyzer (model 7180; Hitachi Corp., Tokyo, Japan).

### 2.6. Micro-CT Analysis

At the end of the treatment, micro-CT (viva CT80; SCANCO Medical, Switzerland) was conducted, and body fat mass was analyzed from the basal region of second lumbar to the distal end of the fifth lumbar.

### 2.7. ELISA Assay

After eight weeks of treatment, carbohydrate levels in collected fecal samples and TG levels in right-lobule liver tissue were measured. The analysis was carried out using a commercial ELISA kit (Biochemical, MI, USA, USA, Triglyceride Assay kit, Asanpharm, Korea, Triglyceride Colorimetric Assay Kit, Cayman, Chemical, USA).

### 2.8. Histopathological Analysis

Left-lobule liver tissue dissected from mice was fixed in 10% neutral-buffered formalin solution. The dehydrated liver tissue was then embedded in paraffin wax, and sections were cut from the paraffin-embedded tissues. These sections stained with Oil-Red-O. Histopathological changes were assessed using an optical microscope (BX53, Olympus, Tokyo, Japan) and image analyzer (Zen 2.3 blue edition; Carl Zeiss, Jena, Germany).

### 2.9. Statistical Analysis

The assumption of homogeneity was tested using Levene’s test. If the overall analysis of variance was significant and the assumption of homogeneity of variance was met, Duncan’s multiple range test was performed. If the assumption of homogeneity of variance was not met, Dunnett’s T3 test was used as the post hoc test. In nonparametric multiple analysis, the Kruskal–Wallis H-test was adopted. If a statistically significant difference was observed between groups, the Mann–Whitney U-test was used to identify the groups. SPSS Statistics 18.0K (IBM Corp., Armonk, NY, USA) was used for all statistical analysis and the level of significance was set at *p* < 0.05.

## 3. Result

In the α-amylase inhibitory activity test, the amount of soluble starch decreased by the enzyme reaction was measured by the iodine reaction method. The inhibitory activity of PM extract against pancreatic α-amylase was determined using different concentrations (1.875, 3.75, 7.5, 15, and 30 mg/mL). As shown in Table 1, the PM extract inhibited the enzyme activities in a dose-dependent way.

Pancreatic lipase is the most important enzyme for the digestion of dietary triacylglycerols. Pancreatic lipase is a key enzyme that hydrolyzes 50% to 70% of total dietary fat in the digestive system, converting TGs to monoglycerides and free fatty acids [18]. Inhibiting pancreatic lipase is an important strategy for treating obesity and other metabolic disorders [19]. In this study, PF extract inhibited pancreatic lipase activity in a concentration-dependent manner, with an Inhibitory concentration (IC_50_) value of 1.68 mg/mL (Table 2). The lipase inhibitory activity of PF extract may be able to suppress dietary fat absorption in vivo as well. 

As a result of in vitro tests, the effect of inhibiting alpha amylase of PM extra and the effect of inhibiting pancreatic lipase of PF extra were the highest at 15 and 5 mg/mL, respectively. Based on the in vitro test results, a DKB-117 extract mixed with a PM:PF = 3:1 (w/w) ratio was prepared to perform in vivo tests by concentration (100, 200, and 300 mg/kg).

The results of the eight weeks of body weight measurement showed that the HFD-treated group had significantly higher values than the ND-treated group, while the DKB-117-300-, orlistat-, and *Garcinia cambogia*–treated groups had significantly decreased weight as compared with the HFD-treated group (*p* < 0.001). The administration of DKB-117-300 was considered to have affected the reduction of body weight in obese models induced by HFD (Figure 1).

The concentrations of TCHO, TGs, and low-density lipoprotein (LDL) in the serum were also remarkably higher in the HFD group compared to those in the normal diet (ND) group, which is also a typical symptom of obesity [20]. 

As shown in Table 3, the HFD induced hyperlipidemia, with increases of plasma triacylglycerol and cholesterol levels. The serum levels of total cholesterol (TCHO), TGs, high-density lipoprotein (HDL), and LDL sharply increased in the HFD group compared to those in the ND group. LDL level was significantly lower (*p* < 0.05) than in the HDF group and reduced by 19.7% after eight weeks of DKB-117-300 administration. No significant alteration was observed in the serum levels of TCHO and TGs. However, these conditions were improved after eight weeks of DKB-117-300 administration (Table 3). These data clearly indicate that DKB-117 intake can effectively mitigate hyperlipidemia induced by HFD by reducing the lipid content in the blood. Furthermore, it was shown that the effects of weight gain reduction by DKB-117-300 treatment had an overall effect on the blood lipid metabolism index.

Sung et al. reported that extracts exhibiting lipase inhibitory activity effectively lower cholesterol and TG levels that were elevated due to HFD intake. In this experiment, the lipase inhibitor (orlistat) effectively lowered TCHO, TGs, and LDL, which are increased by HFD intake [19]. Orlistat is known to inhibit the hydrolysis of dietary fat to free fatty acids so that the fat is not absorbed in the intestine but rather is excreted directly in the feces, thereby reducing a person’s weight and improving their blood lipid levels and glucose metabolism [21]. The DKB-117-300-treated group appeared to present improvements in the lipid metabolism index effectively through lipase inhibitory activity.

Micro-CT scans showed that the total fat, abdominal fat, and subcutaneous fat were all increased in the HFD-induced obese mice compared with those in the ND mice. However, these fats were significantly decreased when DKB-117-100 administration was concomitant. Especially, DKB-117-100 showed an equal or superior effect to that of orlistat and *Garcinia cambogia* used as positive controls. In the case of orlistat, only the visceral fat amount was significantly higher than that in the HFD-treated group (Figure 2).

The experiment was carried out during eight weeks. We measured the weight of inguinal and abdominal fat. After finishing the eight-week test, the weights of inguinal fat and abdominal fat, respectively, were measured after the autopsy. The orlistat group showed a statistically significant decrease in the inguinal fat mass compared to the HFD group, while the DKB-117-100 group showed a statistically significant decrease in the absolute amount of abdominal fat and inguinal fat compared to the HFD group (Figure 3).

As a result of the measurement of fat content in the liver using Oil-Red-O staining, the HFD-treated group revealed a significant increase in fat area in the liver compared to the ND-treated group (*p* < 0.001). In the DKB-117-treated group (100, 200, or 300 mg/kg), the dose-dependent area of fat staining decreased, and the DKB-117-100-treated group showed a significant decrease compared to the HFD-treated group (*p* < 0.05). In addition, the DKB-117-100-treated group showed statistically equivalent efficacy results compared to the *Garcinia cambogia*–treated group (Figure 4).

The liver is an important organ responsible for lipid metabolism, along with fat tissue [22]. Free fatty acids are the most basic elements of the energy metabolism in the body and are transferred to other organs in the form of TGs [23]. An imbalance in lipid metabolism causes an intracellular accumulation of TGs. We confirmed that liver fat was significantly accumulated by eating HFDs. 

As shown in Table 3 and Figure 5, the HFD induced hyperlipidemia, with increases in plasma triacylglycerol and cholesterol levels and induced fatty liver along with an accumulation of triacylglycerol in the liver.

As a result of the analysis of TG content in liver, the HFD-treated group showed a significant increase (*p* < 0.001) as compared with the ND-treated group, while the DKB-117-100-, orlistat- and *Garcinia cambogia*–treated groups showed a significant decrease as compared with the HFD-treated group (*p* < 0.05 or *p* < 0.001). Similar to the results of weight loss, the results here showed that DKB-117-300 intake reduced liver lipid accumulation through a mechanism of digestive enzyme inhibition (Figure 5).

The amount of carbohydrates in the feces was not different between the HFD and ND groups. The DKB-117-100 group showed a significant increase (*p* < 0.01) in carbohydrate content as compared with that in the HFD group, while the orlistat group showed a significant increase in TGs, TCHO, and carbohydrates at the eighth week (Figure 6).

## 4. Discussion

In the present study, the effects of PM and PF extracts on digestive enzymes were assessed. In in vitro studies, PM extract and PF extract have increased dose-dependent inhibitory activity on *α-*amylase (IC_50_ value: 6.13 mg/mL) and pancreatic lipase (IC_50_ value: 1.68 mg/mL), respectively. Notably, α-amylase, one of the digestive enzymes secreted from the pancreas and salivary glands, is involved in important biological processes such as the digestion of carbohydrates. *α*-Amylase inhibitors are well known to be effective in reducing postprandial hyperglycemia by slowing the digestion of carbohydrates and absorbing postprandial glucose. Reducing postprandial hyperglycemia prevents glucose uptake into adipose tissue, inhibiting the synthesis and accumulation of triacylglycerol [24,25].

On the other hand, pancreatic lipase is one of the most important enzymes for the digestion of dietary triacylglycerols. It is well known that dietary lipid is not directly absorbed from the intestines unless it has been subjected to the action of pancreatic lipase [24]. It has been clinically reported that a pancreatic lipase inhibitor, orlistat, prevented obesity and hyperlipidemia through the increment of fat excretion in the feces and the inhibition of pancreatic lipase. Based on this fact, the inhibition of these digestive enzymes is an important factor in the treatment of obesity [26,27]. 

We confirmed the anti-obesity effect of DKB-117 by way of digesting enzyme inhibition through in vivo testing. As a result of the test, DKB-117 extracts revealed effects of reducing weight and total fat in HFD-induced obese C57BL/6J mice. Micro-CT imaging was performed to quantify fat volume (total fat, abdominal fat, and subcutaneous fat). This method can readily discriminate between subcutaneous and abdominal fat [28]. Our results revealed that DKB-117 was able to reduce these fats. The in vivo test results demonstrate that DKB-117 administration regulates serum biochemical parameters (TCHO, TG, HDL, and LDL) in HFD-induced obese mice. These results are associated with an anti-hypercholesterolemic effect of PF [13,14]. In addition, the DKB-117 extracts increased TG and carbohydrate emissions in feces compared to those in the HFD group. It is expected that DKB-117 extracts promote the release of fat and carbohydrates due to the α-amylase inhibitory effect and the lipase inhibitory effect. *Garcinia cambogia,* which is currently used as a dietary supplement, is known to help reduce body fat by inhibiting the enzymes needed to synthesize carbohydrates into fat [29].

On the other hand, the DKB-117 extracts are expected to have an anti-obesity effect through a complex mechanism that promotes carbohydrate release through the inhibition of carbohydrate-degrading enzymes and which inhibits lipid absorption through lipase inhibition. It was observed that DKB-117 can effectively inhibit weight gain in animal experiments through a complex mechanism of inhibition of pancreatic lipase activity and inhibition of amylase activity, which is considered to be a suitable anti-obesity measure for Koreans who use carbohydrate as a staple food.

While the existing research on herbal extract (*Garcinia cambogia, Plantago psyllium, Morus alba*) single-action mechanisms is underway, it has been confirmed through this study that they are effective in anti-obesity in the battle against DKB-117 multifunction machines. Based on the results of in vitro and in vivo tests, the anti-obesity effects of DKB-117 in human application tests will be confirmed. Furthermore, it is determined that active ingredients should be investigated through the study on the separation of components of DKB-117.

In the future, the safety and efficacy of the DKB-117 extracts should be demonstrated through ongoing clinical trials; we plan to search for the activity compound in the DKB-117 extracts as well as perform additional testing of enzyme activity for lipid metabolism so that it can be widely used as a functional food material.

## 5. Conclusions

In summary, results of the present study revealed that DKB-117 extracts possess significant anti-obesity activities. These data clearly show that DKB-117 extracts are expected to have an anti-obesity effect through a complex mechanism that promotes carbohydrate release through the inhibition of carbohydrate-degrading enzymes while blocking lipid absorption through lipase inhibition. Pancreatic lipase is an enzyme that is secreted from the pancreas and hydrolyzes the ester bonds of triglycerides to produce glycerol and fatty acids. 

Decomposed glycerol and fatty acids are absorbed by the mucosal cells of the small intestine and are used as an energy source, but fat that has not been used as an energy source is synthesized into triacylglycerol again through the monoacylglycerol pathway and accumulated in the body. 

Suppressing pancreatic lipase activity inhibits hydrolysis of triacylglycerol into glycerol and fatty acids, thus inhibiting liposuction through the small mucous membrane, reducing the amount accumulated in the body to prevent obesity. 

By suppressing the a-amylase activity, the hydrolysis of ingested polysaccharides is inhibited, so that the absorption of excessively ingested carbohydrates in the body can be reduced and obesity can be prevented. It was confirmed that DKB-117 exhibits an anti-obesity effect by a method in which not only suppression of fat absorption but also suppression of excess carbohydrate absorption act together.

Based on the above results, our team plans to secure the anti-obesity effect and safety of DKB-117 extract through clinical trials.

## Figures and Tables

**Figure 1 nutrients-12-03053-f001:**
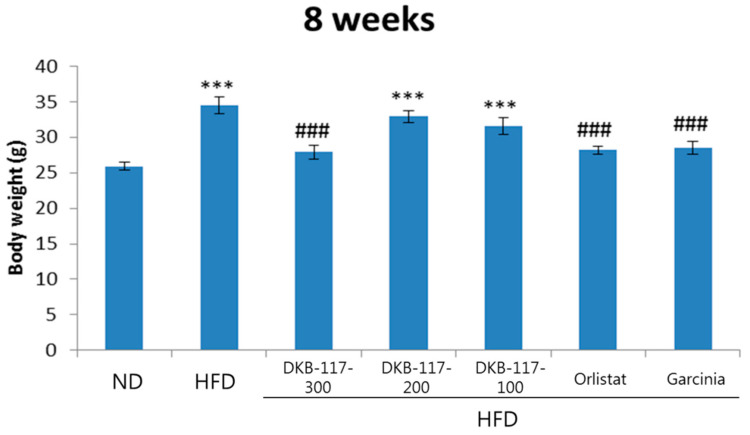
Effects of DKB-117 extracts on body weight in high-fat diet (HFD)–induced obese mice. The values are presented as means ± standard error of the means (SEMs). *** A significant difference at the *p* < 0.001 level was observed versus the normal diet group (ND). ### A significant difference at the *p* < 0.001 level versus the HFD. ND, normal diet control group; HFD, high-fat diet control group; DKB-117-300, DKB-117 300 mg/kg; DKB-117-200, DKB-117 200 mg/kg; DKB-117-100, DKB-117 100 mg/kg; Orlistat, orlistat 40 mg/kg; Garcinia, *Garcinia cambogia* 200 mg/kg.

**Figure 2 nutrients-12-03053-f002:**
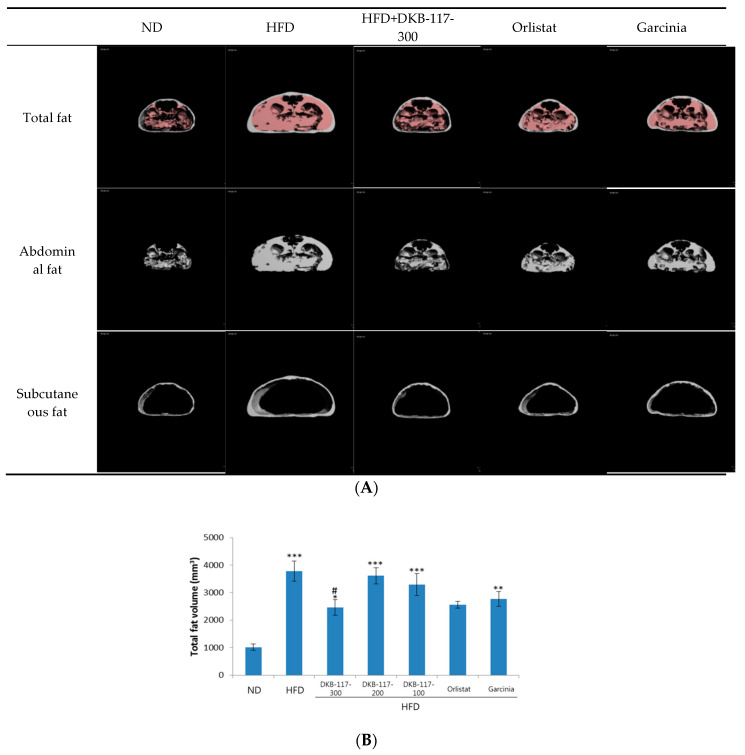
Effects of DKB-117 extracts on fat volume in HFD-induced obese mice. C57BL/6J mice consumed a HFD or ND for eight weeks. (**A**) Transverse microcomputed tomography (micro-CT) images. (**B**) Total fat volume, (**C**) abdominal fat volume, and (**D**) subcutaneous fat volume were measured using micro-CT. The values are presented as means ± SEMs. ***/**/* A significant difference at the *p* < 0.001/*p* < 0.01/*p* < 0.05 level was observed versus the ND. # A significant difference at the *p* < 0.05 level was observed versus the HFD. ND, normal diet control group; HFD, high-fat diet control group; DKB-117-300, DKB-117 300 mg/kg; DKB-117-200, DKB-117 200 mg/kg; DKB-117-100, DKB-117 100 mg/kg; Orlistat, orlistat 40 mg/kg; Garcinia, *Garcinia cambogia* 200 mg/kg.

**Figure 3 nutrients-12-03053-f003:**
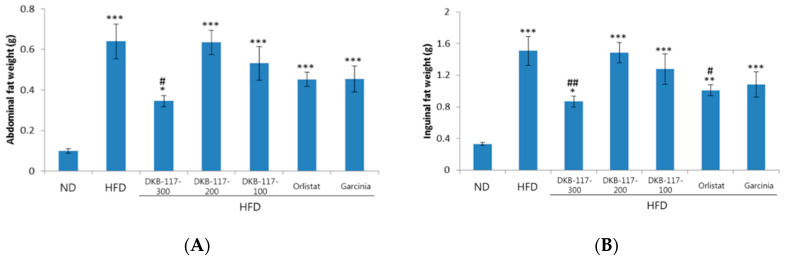
Effect of DKB-117 extracts on (**A**) abdominal and (**B**) inguinal fat weight in HFD-induced obese mice. The values are presented as means ± SEMs. ***/**/* A significant difference at the *p* < 0.001/*p* < 0.01/*p* < 0.05 level versus the ND. ##/# A significant difference at the *p* < 0.01/*p* < 0.05 level versus the HFD. ND, normal diet control group; HFD, high-fat diet control group; DKB-117-300, DKB-117 300 mg/kg; DKB-117-200, DKB-117 200 mg/kg; DKB-117-100, DKB-117 100 mg/kg; Orlistat, orlistat 40 mg/kg; Garcinia, *Garcinia cambogia* 200 mg/kg.

**Figure 4 nutrients-12-03053-f004:**
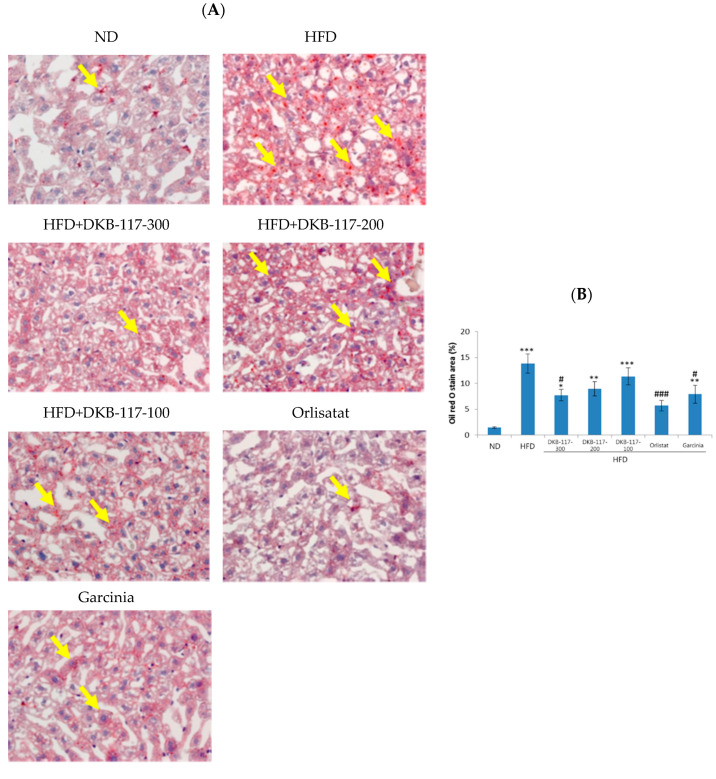
Effect of DKB-117 extracts on lipid accumulation in HFD-induced obese mice. (**A**) The Oil-Red-O staining in liver sections (×400, magnification) and (**B**) quantification of the positively stained area. The values are presented as means ± SEMs. ***/**/* A significant difference at the *p* < 0.001/*p*<0.01/*p*<0.05 level compared to the ND. ###/# A significant difference at the *p* < 0.001/*p* < 0.05 level versus the HFD. ND, normal diet control group; HFD, high-fat diet control group; DKB-117-300, DKB-117 300 mg/kg; DKB-117-200, DKB-117 200 mg/kg; DKB-117-100, DKB-117 100 mg/kg; Orlistat, orlistat 40 mg/kg; Garcinia, *Garcinia cambogia* 200 mg/kg.

**Figure 5 nutrients-12-03053-f005:**
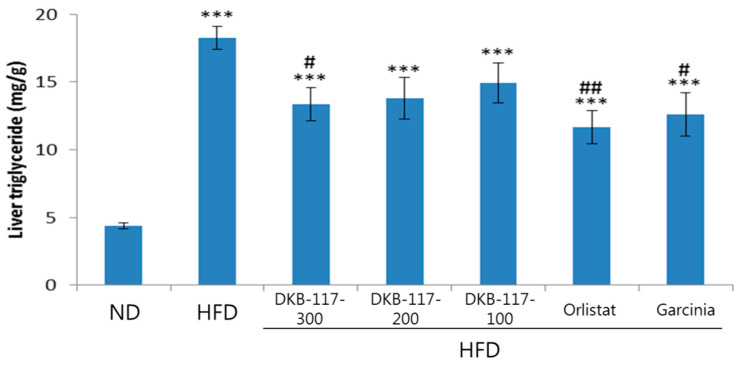
Effect of DKB-117 extracts on liver TG levels in HFD-induced obese mice. The values are presented as means ± SEMs. *** A significant difference at the *p* < 0.001 level versus the ND. ##/# A significant difference at the *p* < 0.01/*p* < 0.05 level versus the HFD. ND, normal diet control group; HFD, high-fat diet control group; DKB-117-300, DKB-117 300 mg/kg; DKB-117-200, DKB-117 200 mg/kg; DKB-117-100, DKB-117 100 mg/kg; Orlistat, orlistat 40 mg/kg; Garcinia, *Garcinia cambogia* 200 mg/kg.

**Figure 6 nutrients-12-03053-f006:**
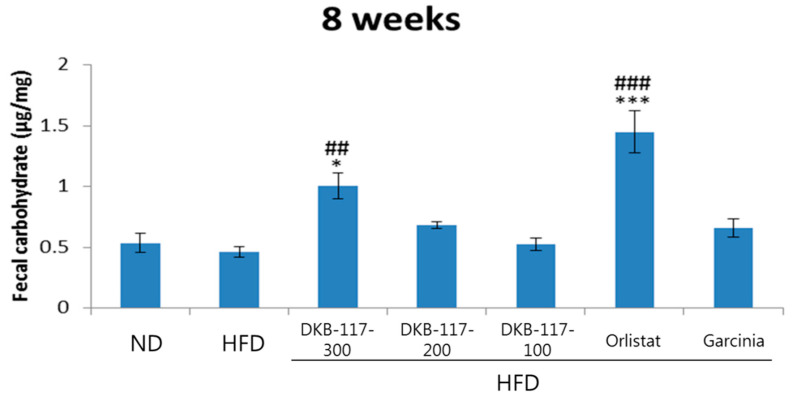
Effects of DKB-117 extracts on feces carbohydrate levels. The values are presented as means ± SEMs. ***/* A significant difference at the *p* < 0.001/*p* < 0.05 level versus the ND. ###/## A significant difference at the *p* < 0.001/*p* < 0.01 level versus the HFD. ND, normal diet control group; HFD, high-fat diet control group; DKB-117-300, DKB-117 300 mg/kg; DKB-117-200, DKB-117 200 mg/kg; DKB-117-100, DKB-117 100 mg/kg; Orlistat, orlistat 40 mg/kg; Garcinia, *Garcinia cambogia* 200 mg/kg.

**Table 1 nutrients-12-03053-t001:** α-Amylase inhibitory activity of PM (*Phaseolus multiflorus* var. albus Bailey) extract.

Sample	Concentration (mg/mL)	Inhibition (%)	Inhibitory Concentration IC_50_ Value (mg/mL)
PM extract	30	77.6 ± 1.36	6.13
	15	82.6 ± 3.20	
	7.5	55.0 ± 3.70	
	3.75	17.6 ± 2.52	
	1.875	5.4 ± 1.10	
Acarbose	1	85.1 ± 1.66	-

Data are presented as medians ± standard deviations (n = 3).

**Table 2 nutrients-12-03053-t002:** Pancreatic lipase (PL) inhibitory activity of PF (*Pleurotus eryngii* var. ferulae) extract.

Sample	Concentration (mg/mL)	Inhibition (%)	IC50 Value (mg/mL)
PF extract	10	74.8 ± 2.29	1.68
	5	76.4 ± 0.85	
	2.5	57.3 ± 1.46	
	1.25	23.9 ± 2.50	
	0.625	4.4 ± 0.76	
Orlistat	1	89.3 ± 0.95	-

Data are presented as medians ± standard deviations (n = 3).

**Table 3 nutrients-12-03053-t003:** Effect of DKB-117 extracts on serum lipid profiles in HFD-induced obese mice after fasting for 15 h at the end of the study.

Experimental Group	Serum Lipid Profiles (mg/dL)			
	TCHO	TG	HDL	LDL
ND	108.6 ± 3.6	60.5 ± 8.5	72.1 ± 1.9	5.2 ± 0.6
HFD	197.6 ± 9.9 ***	148.7 ± 7.5 ***	132.2 ± 4.4 ***	16.1 ± 0.8 ***
HFD+DKB-117-300	173.5 ± 6.9 ***	133.2 ± 7.7 ***	120.2 ± 5.1 ***	12.3 ± 0.6 ***, #
HFD+DKB-117-200	189.7 ± 7.3 ***	128.5 ± 9.5 ***	124.1 ± 5.3 ***	13.9 ± 0.7 ***
HFD+DKB-117-100	195.4 ± 4.7 ***	131.6 ± 9.2 ***	120.6 ± 5.4 ***	14.2 ± 1.1 ***
Orlistat	172.5 ± 5.4 ***	133.9 ± 10.8 ***	117.6 ± 2.6 ***	12.5 ± 0.9 ***, #

The values are presented as means ± SEMs. *** A significant difference at the *p* < 0.001 level was observed versus the ND. # A significant difference at the *p* < 0.05 level was observed versus the HFD. ND, normal diet control group; HFD, high-fat diet control group; DKB-117-300, DKB-117 300 mg/kg; DKB-117-200, DKB-117 200 mg/kg; DKB-117-100, DKB-117 100 mg/kg; Orlistat, orlistat 40 mg/kg; Garcinia, *Garcinia cambogia* 200 mg/kg, TG: triglyceride; HDL, high-density lipoprotein, LDL: low-density lipoprotein, TCHO: total cholesterol.

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
