# Peer review of "Anti-Obesity Effect of DKB-117 through the Inhibition of Pancreatic Lipase and α-Amylase Activity"

_nutrients, 2020, doi:10.3390/nu12103053_

Round 1
Reviewer 1 Report
In this work the authors investigate the effect of combined plants products (Phaseolus multiflorus var. albus Bailey extract and Pleurotus eryngii var. ferulae extract) on alpha amylase and pancreatic lipase activity. They show that 300mg/kg of this dietary supplement can improve body composition with decreased body fat mass. This kind of study are important to add nutritional opportunities to manage weight and metabolic status. However, there are some important concerns that the authors should address.
Abstract line 17: what is PE is it the same that PF ?
Introduction line 67: please insert reference number for Wang et al and Alam and al.
Line 87 same that previously asked what is PE extract ? do the authors mean PF extract ?
Line 86-93 the authors write about a clinical trial using DKB-117. It not clear if DKB-117 is or is different from the “dietary supplement ingredient” mentioned line 76. As the protocol presented in the method part is an animal study, paragraph line 86-93 appear confusing in the manuscript and is not appropriate.
Result: figure 3 and 4 orlistat and garcinia result do not appear on the graph.
The authors show that there is an increase in carbohydrate in fecal sample. However if the product is to prevent hyperglycemia this parameter should be added to the manuscript. As the authors in the manuscript argue reduction of hyperglycemia it would be relevant to show the effect of the plant extract on basal glucose level and/or glucose OGTT or IPGTT response.
Author Response
Thanks for the good comment.
The revision was completed by reflecting the part you said.
Please checking details have been attached as a file.
Response to Reviewer 1 Comments
In this work the authors investigate the effect of combined plants products (Phaseolus multiflorus var. albus Bailey extract and Pleurotus eryngii var. ferulae extract) on alpha amylase and pancreatic lipase activity. They show that 300mg/kg of this dietary supplement can improve body composition with decreased body fat mass. This kind of study are important to add nutritional opportunities to manage weight and metabolic status. However, there are some important concerns that the authors should address.
Point 1: Abstract line 17: what is PE is it the same that PF ?
Response 1:
PE and PF are the same extract. Edited according to reviewer's opinion.
Point 2: Introduction line 67: please insert reference number for Wang et al and Alam and al.
Response 2:
We have inserted a reference number according to the reviewer's opinion.
Point 3: Line 87 same that previously asked what is PE extract ? do the authors mean PF extract ?
Response 3:
PE and PF are the same extract. Edited according to reviewer's opinion.
Point 4: Line 86-93 the authors write about a clinical trial using DKB-117. It not clear if DKB-117 is or is different from the “dietary supplement ingredient” mentioned line 76. As the protocol presented in the method part is an animal study, paragraph line 86-93 appear confusing in the manuscript and is not appropriate.
Response 4:
Lines 86-93 have been deleted according to the reviewer's opinion.
Point 5: figure 3 and 4 orlistat and garcinia result do not appear on the graph.
Response 5:
Edited according to reviewer's opinion.
Point 6: The authors show that there is an increase in carbohydrate in fecal sample. However if the product is to prevent hyperglycemia this parameter should be added to the manuscript. As the authors in the manuscript argue reduction of hyperglycemia it would be relevant to show the effect of the plant extract on basal glucose level and/or glucose OGTT or IPGTT response.
Response 6:
Thanks for the comments.
This study is an experiment confirming that the decomposition of carbohydrates into monosaccharides is reduced by inhibiting the activity of alpha-amylase intestinal digestive enzymes, and thus carbohydrates are increased in fecal samples, and does not claim to decrease hyperglycemia. In this study, as you mentioned, there is a lack of scientific evidence for hyperglycemia, so we will plan to proceed with OGTT and IPGTT evaluation in the future.
Reviewer 2 Report
The authors have performed a study to validate the anti-obesity effect by DKB-117 (PM and PF extract), concerning the inhibition of digestive enzyme. In vitro and in vivo design has been proposed to answer the question for whether PM and PF extract have inhibitory activities against a-amylase and pancreatic lipase and how DKB-117 mixture shows anti-obesity. The paper is well written and highlights to find a natural compounds for obesity treatment.
Specific points
#1. In introduction, authors claimed the issue for heart attack or stoke by existing drug and it will be a good reason to find a natural compounds. However, I cannot find any supporting data or discussion for this issue. If authors have data for that, please inset it in the manuscript or they should clarify this point in discussion part.
#2. On line 78 (In the literature,...), please update the reference.
#3. In conclusion, line 348 says a complex mechanism. I think that authors should explain it with more details (which kind of mechanism is involved in...).
#4. In methods part, please add missing information about chemical or instrument (which company...).
Author Response
Thanks for the good comment.
The revision was completed by reflecting the part you said.
Please checking details have been attached as a file.
Response to Reviewer 2 Comments
The authors have performed a study to validate the anti-obesity effect by DKB-117 (PM and PF extract), concerning the inhibition of digestive enzyme. In vitro and in vivo design has been proposed to answer the question for whether PM and PF extract have inhibitory activities against a-amylase and pancreatic lipase and how DKB-117 mixture shows anti-obesity. The paper is well written and highlights to find a natural compounds for obesity treatment.
Point 1: In introduction, authors claimed the issue for heart attack or stoke by existing drug and it will be a good reason to find a natural compounds. However, I cannot find any supporting data or discussion for this issue. If authors have data for that, please inset it in the manuscript or they should clarify this point in discussion part.
Response 1:
To confirm the safety of DKB-117, we plan to secure safety through clinical trial study and GLP toxicity tests. This is mentioned on line 345.
Point 2: On line 78 (In the literature,...), please update the reference.
Response 2:
We have inserted a reference number according to the reviewer's opinion.
Point 3: In conclusion, line 348 says a complex mechanism. I think that authors should explain it with more details (which kind of mechanism is involved in...).
Response 3:
Edited according to reviewer's opinion (line 367).
Pancreatic lipase is an enzyme that is secreted from the pancreas and hydrolyzes the ester bonds of triglycerides to produce glycerol and fatty acids.
Decomposed glycerol and fatty acids are absorbed by the mucosal cells of the small intestine and are used as an energy source, but fat that has not been used as an energy source is synthesized into triacylglycerol again through the monoacylglycerol pathway and accumulated in the body.
Suppressing pancreatic lipase activity inhibits hydrolysis of triacylglycerol into glycerol and fatty acids, thus inhibiting liposuction through the small mucous membrane, reducing the amount accumulated in the body to prevent obesity.
By suppressing the a-amylase activity, the hydrolysis of ingested polysaccharides is inhibited, so that the absorption of excessively ingested carbohydrates in the body can be reduced and obesity can be prevented. It was confirmed that DKB-117 exhibits an anti-obesity effect by a method in which not only suppression of fat absorption but also suppression of excess carbohydrate absorption act together.
Point 4: In methods part, please add missing information about chemical or instrument (which company...).
Response 4:
Edited according to reviewer's opinion(line 115 and line 124).
Reviewer 3 Report
I was honored to review the manuscript entitled “Anti-obesity Effect of DKB-117 through the 3 Inhibition of Pancreatic Lipase and α-amylase 4 Activity” submitted to Nutrients. The study presents high quality and deals with the important clinical issue, such type of study is needed. I have only few small remarks that authors should address properly.
There are only some points to correct:
- please provide the list of abbreviations
- whether the study was approved by the local ethics committee? please provide the number of ethical approval
- introduction and discussion section need improvement – please provide information on how your results will translate into clinical practice
- in the discussion section please provide study strong points and study limitation section
- please correct typos
Author Response
Thanks for the good comment.
The revision was completed by reflecting the part you said.
Please checking details have been attached as a file.
Response to Reviewer 3 Comments
I was honored to review the manuscript entitled “Anti-obesity Effect of DKB-117 through the 3 Inhibition of Pancreatic Lipase and α-amylase 4 Activity” submitted to Nutrients. The study presents high quality and deals with the important clinical issue, such type of study is needed. I have only few small remarks that authors should address properly.
Point 1: please provide the list of abbreviations
Response 1:
Edited according to reviewer's opinion.
Point 2: whether the study was approved by the local ethics committee? please provide the number of ethical approval
Response 2:
The study has been approved by the local ethics committee, and the approval numbers are HKS2018-05-018 and 2018AS0166.
Point 3: introduction and discussion section need improvement – please provide information on how your results will translate into clinical practice
Response 3:
[Introduction : line on 98]
Research is actively underway to find effective anti-obesity drugs or anti-obesity health functions. This researcher wants to confirm the anti-obesity effect by mixing Phaseolus multiflorus var. albus Bailey with α-amylase inhibitory effect and Pleurotus eryngii var. ferulae reported to have a pancreatic lipase inhibitory effect. In this study, the optimum mixing ratio of PF extract and PF extract was selected by the In vitro test, and we sought to determine the anti-obesity effect of DKB-117 in digestive enzyme inhibition in a mouse model of obesity induced by HFD.
[discussion : line on 348 ~ 357 ]
While the existing research on herbal extracts (Garcinia cambogia, Plantago psyllium, Morus alba) single action mechanisms is underway, it has been confirmed through this study that they are effective in anti-obesity in the battle against DKB-117 multifunction machines. Based on the results of in vitro and in vivo tests, the anti-obesity effects of DKB-117 in human application tests will be confirmed. Furthermore, it is determined that active ingredients should be investigated through the study on the separation of components of DKB-117 .
Point 4: in the discussion section please provide study strong points and study limitation section
Response 4:
Edited according to reviewer's opinion.
It was observed that DKB117 can effectively inhibit weight gain in animal experiments through a complex mechanism of inhibition of pancreatic Lipase activity and inhibition of amylase activity, which is considered to be a suitable anti-obesity measure for Koreans who use carbohydrate as a staple food.
Point 5: please correct typos
Response 5:
Edited according to reviewer's opinion (line on 17).
Round 2
Reviewer 1 Report
no other comment